# Modest Reduction in CAG Repeat Length Rescues Motor Deficits but Not Purkinje Cell Pathology and Gliosis in Spinocerebellar Ataxia Type 1 Mice

Stephen Gilliat , Juao-Guilherme Rosa , Genevieve Benjamin, Kaelin Sbrocco, Wensheng Lin and Marija Cvetanovic *

Department of Neuroscience, Institute for Translational Neuroscience, University of Minnesota, 2101 6th Street SE, Minneapolis, MN 55455, USA
* Correspondence: mcvetano@umn.edu; Tel.: +1-612-626-4918

**Abstract:** Spinocerebellar ataxia type 1 (SCA1) is a fatal, dominantly inherited neurodegenerative disease caused by the expansion of CAG repeats in the *Ataxin-1* (*ATXN1*) gene. SCA1 is characterized by the early and prominent pathology of the cerebellar Purkinje cells that results in balance and coordination deficits. We previously demonstrated that cerebellar astrocytes contribute to SCA1 pathogenesis in a biphasic, stage of disease-dependent manner. We found that pro-inflammatory transcriptional regulator nuclear factor κ-light-chain-enhancer of activated B cells (NF-κB) signaling in astrocytes has a neuroprotective role during early-stage SCA1. Here, we sought to examine whether further inducing NF-κB activation in astrocytes of SCA1 model mice at an early stage of the disease has therapeutic benefits. To perform this task, we created a novel Slc1a3-CreER$^T$/IKKβ$_{CA}$/*ATXN1[82Q]* triple transgenic mouse model in which TMX injection at 4 weeks of age results in the expression of constitutively active inhibitor of kB kinase beta (IKKβ$_{CA}$), the main activator of NF-κB signaling. As we evaluated SCA1-like phenotypes, we noticed that *ATXN1[82Q]* mice did not exhibit motor deficits anymore, even at very late stages of the disease. We sequenced the mutant *ATXN1* gene and discovered that the CAG repeat number had decreased from 82 to 71. However, despite the loss of motor phenotype, other well-characterized SCA1-changes, including atrophy of Purkinje cell dendrites, hallmarks of cerebellar astrogliosis and microgliosis, and Purkinje cell disease-associated gene expression changes, were still detectable in *ATXN1[71Q]* mice. We found delayed PC atrophy and calbindin reduction in SCA1 mice expressing IKKβ$_{CA}$ in astrocytes implicating beneficial effects of increased NF-κB signaling on Purkinje cell pathology. The change in the motor phenotype of SCA1 mice with CAG reduction prevented us from evaluating the neuroprotective potential of IKKβ$_{CA}$ on motor deficits in these mice.

**Keywords:** ATAXIN-1; astroglia; neuroprotective; nuclear factor kappa b; cerebellum; neurodegeneration



## 1. Introduction

Spinocerebellar ataxia type 1 (SCA1) is a fatal genetic neurodegenerative disease with an autosomal dominant pattern of inheritance [1]. Like Huntington's disease and other types of spinocerebellar ataxia (SCA 2, 3, 6, 7, and 17), SCA1 is classified as a polyglutamine neurodegenerative disease because it is caused by an increased number of glutamine codon CAG repeats in the coding region of the gene *ATAXIN1* (*ATXN1*) [2,3]. While the normal version of human *ATXN1* features 4–39 CAG repeats with CAT interruptions, the disease-causing version of the gene can contain 39–83 pure repeats [4,5]. A greater number of CAG repeats corresponds to an earlier onset of the disease, as individuals with 40–50 repeats develop SCA1 symptoms in midlife, while individuals with 70 or more repeats develop symptoms in childhood [6]. Additionally, a higher count of CAG repeats is correlated with greater disease severity [5]. SCA1 is also characterized by anticipation, as the CAG repeat

expansion is unstable and can grow longer in subsequent generations, leading to an earlier age of SCA1 onset and increased severity of disease in offspring [7].

While the typical age of SCA1 onset is usually in the patient's thirties, disease symptoms can appear between 4 and 74 years of age [8]. SCA1 patients experience progressively worsening motor symptoms, including, but not limited to, a wide, unsteady gait, slowing and slurring of speech, oculomotor abnormalities such as nystagmus, and dysphagia [9,10]. Patients can also show signs of cognitive dysfunction, such as increased emotional lability. Progression from disease onset to death usually takes 10–20 years [11,12]. The most common cause of death in SCA1 patients is pneumonia due to restrictive respiratory failure. As of yet, no cure exists for SCA1. Though various pharmacological treatments have been tested for symptom alleviation, they have shown mixed efficacy across patient populations.

Despite the widespread expression of mutant ATXN1 within the CNS, SCA1 is pathologically characterized by the early and prominent loss of Purkinje cell neurons in the cerebellum [13–15]. One possible explanation for the particular vulnerability of Purkinje cells in SCA1 pathology is that the mutant ATXN1 disrupts cellular and molecular pathways that are specific to these cells and necessary for their proper functioning [1]. Another potential contributing factor to Purkinje cell dysfunction and loss is altered astrocyte activity, particularly that of Bergmann glia, which associate closely with Purkinje cells [16]. In the cerebellum, there are three types of astrocytes: Bergmann glia that span the molecular and Purkinje cell layers of the cerebellar cortex, velate astrocytes in the granular layer of the cerebellar cortex, and fibrous astrocytes in the white matter [16]. In particular, Bergmann glia are closely associated with Purkinje cells in the cerebellum, with most synapses on Purkinje cell dendrites being enveloped by Bergmann glia processes [17]. Bergmann glia provide many crucial supports to Purkinje cells, including maintenance of Purkinje cell structure and function and regulation of Purkinje cell membrane potential and activity [18]. Dysfunction of Bergmann glia can have disastrous results for Purkinje cells, as alterations in Bergmann glia calcium signaling can disrupt Purkinje cell firing rate [18], and ablation of Bergmann glia can lead to Purkinje cell dendrite degeneration along with motor impairment [17].

Glia react to central nervous system damage caused by an acute injury or chronic disease through a process called reactive gliosis [19,20]. Hallmarks of reactive astrogliosis include hypertrophy of astrocyte cell body and cellular processes as well as upregulation of glial fibrillary acidic protein (GFAP) [21,22]. Reactive astrocytes can release many different types of molecules, ranging from cytokines to neurotrophic factors [23]. Reactive astrogliosis is known to play a role in many neurodegenerative diseases, including Alzheimer's disease, Parkinson's disease, Huntington's disease, and amyotrophic lateral sclerosis (ALS) [24–29]. Reactive astrocytes show disruption of normal astrocyte homeostatic functions such as the maintenance of glutamate and potassium ion concentrations, and this disruption exacerbates the pathology of neurodegenerative disease [23,29,30].

Reactivity of astrocytes and microglia was shown to occur in SCA1 [31,32]. Astrogliosis and microgliosis have both been noted in post-mortem analyses of human ataxia patients [16]. Reactivity of Bergmann glia was seen in one of the first mouse models of SCA1, *ATXN1[82Q]* mice that express a mutant *ATXN1* with 82 CAG repeats only in Purkinje cells [33]. In these mice, gliosis was observed in the post-symptomatic phase of the disease, alongside Purkinje cell loss. However, recent studies showed that gliosis begins early in SCA1 progression in *ATXN1[82Q]* mice, where hallmarks of Bergmann astrogliosis and microgliosis can be detected in PC and molecular layers as early as 4 weeks of age [31].

In our previous study, we examined the role of nuclear factor κB (NF-κB) in cerebellar astrocytes on SCA1 pathogenesis [34]. As one of the key transcriptional factors implicated in neuroinflammation, NF-κB is activated in many neurodegenerative disorders [35–37]. Recent studies suggest that NF-κB is an important regulator of gene expression in human astrocytes under both physiological and inflammatory conditions [35,38,39]. Under normal conditions, most of NF-κB is cytoplasmically sequestered and kept inactive by the inhibitor of nuclear factor κB protein (IκB) proteins [40]. During inflammation, NF-κB is canonically

activated following phosphorylation of the catalytic unit of IκB kinase (IKK) complex (IKKβ) at Ser180 [41,42]. Activated IKKβ, in turn, phosphorylates IκB proteins to mark them for ubiquitination and degradation, and allows for the release of NF-κB into the nucleus [43–46].

We previously characterized NF-κB's role at two stages of SCA1 disease progression through a novel SCA1 mouse model enabling cell type and temporal control of IKKβ conditional depletion [34]. As these previous results suggested that astroglial NF-κB signaling may exert a novel neuroprotective role early in the disease, here we aimed to explore the therapeutic benefits of further activating NF-κB signaling early in SCA1.

## 2. Methods and Materials

### 2.1. Generation of ATXN1[82Q];IKKβ$_{CA}$;Slc1a3-CreER$^T$ Mice

All of the mouse lines are on C57BL/6 background. Creation and the genetic background of the *ATXN1[82Q]* mice have been previously described (Burright et al., 1995). While these mice were originally generated in an FVB/N background, we backcrossed them into C57BL/6 mice for 10 generations. In this mouse model, the expression of mutant *ATXN1* with an expanded 82 CAG repeat in its coding sequence is driven by a *Purkinje cell-specific 2(Pcp2/L7)* promoter. We received these mice as a gift from Harry Orr. In order to achieve selective activation of astroglial NF-κB signaling, *ATXN1[82Q]* mice were crossed with *R26Stop$^{FL}$IKKβ$_{CA}$* mice on C57BL/6 background, received as a gift from Wensheng Lin (Lei et al., 2020). These mice feature a constitutively active form of IKKβ, in which activation is mimicked through two serine to glutamine substitutions in the activation loop of the kinase domain [47]. Expression of the IKKβ$_{CA}$ transgene, its FLAG tag, and a GFP reporter is blocked by a floxed STOP cassette (Lei et al., 2020). When bred to mice expressing tamoxifen-activated CreER$^T$ recombinase under the control of the GLAST/Slc1a3 promoter (Jackson laboratory, #012586, Tg(Slc1a3-cre/ERT)1Nat/J) [34], following tamoxifen injection, the floxed STOP cassette is deleted, and astrocyte-specific IKKβ$_{CA}$ expression is achieved.

*Slc1a3-cre/ER$^T$* mice on C57BL/6 background were obtained from the Jackson laboratory (#012586; Tg(Slc1a3-cre/ERT)1Nat/J). We used an equal number of mice of both sexes. Mice were randomly allocated to TMX-treated groups (i.e., to be injected with a synthetic agonist tamoxifen (TMX) to activate the Cre recombinase and thus induce the expression of IKKβ$_{CA}$) or oil–injected control group. We injected *ATXN1[82Q];IKKβ$_{CA}$;Slc1a3-cre/ER$^T$* mice with TMX at 4 weeks of age for the early disease stage time point. At 4 weeks of age, the mutant ATAXIN1 is actively being expressed in Purkinje neurons (from postnatal day 10), the astrogliosis already initiated (at 3 weeks), but there is no detectable change in the motor behavior of *ATXN1[82Q]* mice (i.e., the mice were at the pre-symptomatic stage) [31,48,49]. In all the experiments on these mice, investigators were blinded to the genotype/treatment until data were fully collected.

Animal experimentation was approved by the University of Minnesota and was conducted in accordance with the National Institutes of Health's Principles of Laboratory Animal Care (86–23, revised 1985), and the American Physiological Society's Guiding Principles in the Use of Animals.

### 2.2. Repeat Sequencing

To determine the size of the CAG repeat expansion in *ATXN1[82Q]* mice, we used 5EX2B (AGG TTC ACC GGA CCA GGA AGG), Ruby (AAT GAA CTG GAA GGT GTG CGG C), Gap60-1 (AAC TTT GGC ATT GTG GAA GG), and Gap60-2 (ACA CAT TGG GGG TAG GAA CA), ran a 1% agarose gel containing our samples and excised the appropriate band. We then used a PureLink™ Quick Gel Extraction Kit (Thermo Fisher Scientific, Waltham, MA, USA) to extract the DNA and sent the purified DNA to the University of Minnesota Genomics center to be sequenced. Returned files were analyzed using FinchTV or SnapGene Viewer 6.0 software.

## 2.3. Rotarod Analysis

An accelerating rotating rod test allowed us to evaluate coordination and motor skill acquisition (#47600; Ugo Basile, Gemonio, Italy). Four-month-old animals were placed on the rod (3 cm diameter, 5.7 cm lane, 16 cm height to fall) and subjected to four trials per day for a period of 4 days. Each trial lasted for 10 min maximum. Rotarod paradigm consisted of acceleration from 5 to 40 rpm over minutes 0 to 5, followed by 40 rpm constant speed from 5 to 10 min. Latency to fall was recorded. Additionally, two consecutive rotation events, if separated by a span of 10 s, were also considered a fall. Mice were given 10 min of rest between trials. Behavioral scores were subject to statistical analysis using one-way ANOVA with post hoc Bonferroni testing.

## 2.4. Immunofluorescent (IF) Staining

Mouse brains were fixed overnight in 4% paraformaldehyde, cryoprotected by immersing in 30% sucrose, stored in OCT at $-80$ °C and then sectioned on cryostat (Leica, CM 1850, Wetzlar, Germany) into 40 µm sections. The sections were washed in phosphate-buffered saline (PBS) and incubated in blocking buffer (5% normal goat serum in 1% PBS-Triton X) for 1 h at room temperature. The tissues were then incubated in the blocking buffer overnight at room temperature containing appropriate dilutions of primary antibodies (Calbindin #13176S, Cell Signaling Technology; Iba-1 #019-19741, Wako; GFAP #AB5541, EMD Millipore Corporation (Burlington, MA, USA); VGLUT2 #AB2251-I, EMD Millipore Corporation). The tissues were washed three times for 5 min each in PBS, and incubated for 3 h at room temperature in appropriate secondary antibodies conjugated with Alexa fluorophores (Alexa Fluor® Dyes; Life technologies (Carlsbad, CA, USA)) diluted in blocking buffer (1:400). Following three PBS washes for 5 min each, the tissues were mounted on slides with Vectashield Hardset mounting medium containing 4′,6′-diamino-2-phenylindole (DAPI) (#H-1500; Vector Laboratories, Burlingame, CA, USA) for observation under the microscope (Olympus FluoView™ FV1000, Tokyo, Japan). At least six different z-stacks of 20 µm from each mouse were studied, and at least 3 mice from each genotype and treatment were examined. All images were taken at 20× magnification.

## 2.5. Quantitative Analysis of Immunofluorescent Staining

Quantitative analysis of immunofluorescent-stained tissues was performed by using ImageJ (National Institutes of Health). The intensity of staining for calbindin and GFAP was determined by measuring mean gray value in the molecular and Purkinje cell layers of lobule VI of the cerebellum. Microglia density was determined by dividing the number of Iba-1-positive microglia found within the molecular and Purkinje cell layers of lobule VI of cerebellar cortex by the region of interest (ROI) area. Cerebellar molecular layer thickness was determined as the distance from the base of the Purkinje cell bodies to the end of the dendrite (as marked by calbindin staining). Minimum of three primary fissures from each mouse and six to eight measurements per primary fissure were measured. For assessing the height of climbing fibers extending along Purkinje neuron dendrites, the distance from the Purkinje neuron soma to the end of VGLUT2 staining was measured, and the extension of climbing fibers was depicted relative to the molecular layer thickness (measured as described with calbindin staining). Data were analyzed using one-way ANOVA with Bonferroni's multiple comparison post hoc tests.

## 2.6. RT-qPCR

Total RNA was extracted from dissected mouse cerebella using TRIzol (Life Technologies, Waltham, MA, USA) according to the manufacturer's instructions. RNA was then treated with DNase to remove any contaminating genomic DNA (TURBO DNA-free™ Kit #AM1907, Thermo Scientific, Waltham, MA, USA), and reverse transcribed in independent duplicate reactions using random hexamers and SuperScript® III First-Strand Synthesis System (#18080051, Thermofisher Scientific, Waltham, MA, USA). Quantitative PCR was performed in a Light Cycler®480 II (Roche, Basel, Switzerland) by using SYBR Green Mas-

ter Mix (Roche, Basel, Switzerland) and appropriate primers (IDT Primetime or custom primers summarized in Table 1) that target genes abundantly expressed in Purkinje neurons which correlate with disease progression (identified as 'Magenta cluster of genes' from Ingram et al., 2016). The mRNA levels were determined with $2^{-\Delta\Delta Ct}$ (Ct = threshold cycle) formula normalized to 18S RNA using wild-type mice as a reference. Data were analyzed using one-way ANOVA.

**Table 1.** Primers used for RT-qPCR. Table details the gene group for each primer, type of primer (PrimeTime or custom), and sequence (forward and reverse where applicable).

| Primer Name | Gene Group | PrimeTime or Custom | Forward Sequence (5′-3′) | Reverse Sequence (5′-3′) |
|---|---|---|---|---|
| Calbindin | Magenta | Custom | AAG-GCT-TTT-GAG-TTA-TAT-GAT-CAG | TTC-TTC-TCA-CAC-AGA-TCT-TTC-AGC |
| PCP4 | Magenta | Custom | CCA-ACG-GAA-AAG-ACA-AGA-CG | TGT-CGA-TAT-CAA-ATT-CTT-CTT-GGA |
| Homer3 | Magenta | Custom | TGA-AGA-AGA-TGC-TGT-CAG-AAG-G | CTG-TCC-TGA-AGC-GCG-AAG |
| RGS8 | Magenta | Custom | CTG-TCA-CAC-AAA-TCA-GAC-TCC-TG | TGC-TTC-TTC-CGT-GGA-GAG-TC |
| ITPR | Magenta | Custom | GAA-GGC-ATC-TTT-GGA-GGA-AGT | ACC-CTG-AGG-AAG-GTT-CTG-C |
| Inpp5a | Magenta | Custom | ATT-CGG-ACA-CTT-TGG-AGA-GC | CCT-TTT-CTT-GAC-CAT-TTG-CAC |
| Garnl3 | Magenta | Custom | TCA-TGA-AGC-CGT-GTG-TGC | CAG-GGA-TGG-GAG-GTC-ATC |
| 18S rRNA | Reference gene (used in all experiments) | Custom | AGT-CCC-TGC-CCT-TTG-TAC-ACA | CGA-TCC-GAG-GGC-CTC-ACT-A |

*2.7. Statistical Analysis*

Wherever possible, sample sizes are calculated using power analyses based on the standard deviations from our previous studies, with a significance level of 5% and power of 90%. Statistical tests were performed with GraphPad Prism. For rotarod, IHC, and RT-qPCR, we used one-way ANOVA followed by Bonferroni post hoc tests, and when there were only two groups to compare, we used a two-tailed Student's *t*-test.

*2.8. Data Availability*

The data that support the findings of this study are available from the corresponding author upon reasonable request.

**3. Results**

*3.1. Creation of Mice with Conditional, Astroglia-Selective, and TMX-Dependent Expression of Constitutively Active IKKβ*

The transgenic *ATXN1[82Q]* mice express mutant *ATXN1[82Q]* protein selectively in Purkinje neurons and are one of the best-characterized mouse models of SCA1 [33]. *ATXN1[82Q]* mice closely mirror the cerebellar aspects of the human disease at cellular and behavioral levels [6]. Motor coordination deficits and dendritic atrophy of Purkinje neurons begin around 12 weeks of age in *ATXN1[82Q]* mice (Supplementary Figure S1) [32,50–52], with loss of Purkinje neurons following around 24 weeks [50,53,54]. We previously demonstrated that cerebellar reactive astrogliosis is present in *ATXN1[82Q]* mice at three weeks of age and thus precedes motor deficits [31].

To characterize whether further activation of astroglial NF-κB early in disease ameliorates SCA1 pathogenesis, we created a novel conditional triple transgenic SCA1 mouse

*ATXN1[82Q];IKKβ $_{CA}$;Slc1a3-cre/ER$^T$* line (Figure 1). These mice allow for selective activation of classical NF-κB signaling in SCA1 astroglia via TMX injection. To create these transgenic mice, three separate mouse lines were crossed: (1) a conditional *IKKβ$_{CA}$* mouse line (*R26Stop$^{FL}$IKKβ$_{CA}$* mice, expression of a constitutively active form of IKKβ, in which activation is mimicked through two serine to glutamine substitutions in the activation loop of the kinase domain [47], is blocked by a floxed STOP cassette [55]; (2) a *Slc1a3-cre/ER$^T$* mouse line that utilizes the astroglial-specific promoter *solute carrier family 1 member 3(Slc1a3)* to drive expression of tamoxifen (TMX)-inducible Cre recombinase [56,57]; and (3) the mutant *ATXN1[82Q]* mouse line that selectively expresses mutant ATXN1 under the Purkinje neuron-specific promoter *PCP2* (Burright et al., 1995). This system allows for temporal control of astroglial NF-κB activity. Following intraperitoneal (lP) injections of TMX, astroglia-specific CreER$^T$ recombinase becomes functional and deletes floxed STOP cassette, which leads to astroglial expression of IKKβ$_{CA}$ and ultimately to NF-κB activation in *IKKβ$_{CA}$;Slc1a3-cre/ER$^T$* and *ATXN1[82Q];IKKβ$_{CA}$;Slc1a3-cre/ER$^T$* lines (from now on for simplicity labeled *IKKβ$_{CA}$* and *ATXN1[82Q];IKKβ$_{CA}$*, respectively) (Figure 1A). We validated astroglial selectivity of cre recombination in the cerebellum of *Slc1a3-cre/ER$^T$* mouse line using a reporter line that shows the overlap of cre recombination-induced fluorescence with S100β, a marker of astroglia (Supplementary Figure S2).

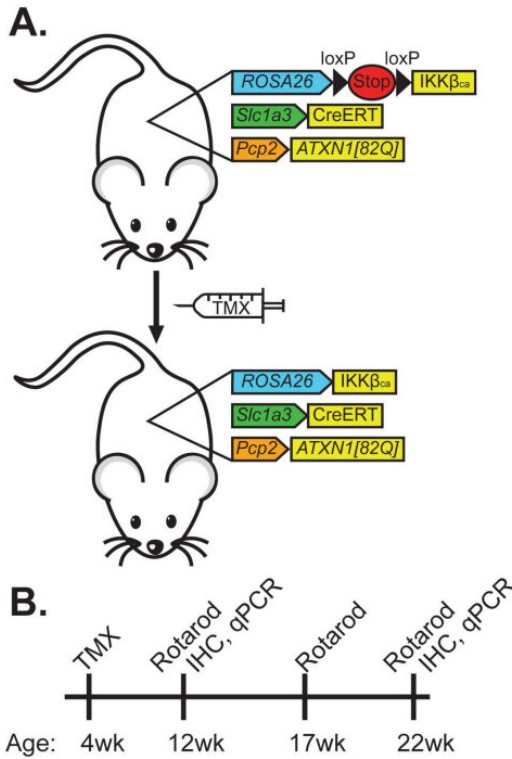

**Figure 1.** Generation of the novel SCA1 transgenic mouse line to induce astroglial NF-κB signaling in a TMX-dependent manner. (**A**) Schematics illustrate that intraperitoneal injection tamoxifen (TMX) activates Cre-recombinase (CreERT) expressed under astrocyte-specific *Slc1a3* promoter, allowing for removal of loxP enclosed STOP cassette and expression of constitutively active IKKβ (IKKβCA) expression under control of the endogenous *ROSA26* promoter. Animals also express mutant *ATXN1*[82Q] under Purkinje cell-specific promoter *Pcp2*. (**B**) Timeline schematic showing animal age (bottom) and experiment (top). Mice were injected with TMX or oil control at four weeks of age, and rotarod was performed at 12, 17, and 22 weeks of age. Animals were sacrificed after rotarod experiments occurring at 12 or 22 weeks of age. Tissue was analyzed by immunohistochemistry (IHC) or RT-qPCR.

To induce expression of astroglial IKK$_{CA}$ early in the disease, mice were injected with TMX at 4 weeks of age. We examined the effects on motor deficits throughout disease progression by performing rotarod at 12, 17, and 22 weeks of age. Using immunohistochemistry and RT-qPCR, we examined cerebellar pathology at two stages: mid (12 weeks) and late stage (after 22 weeks) (Figure 1B).

### 3.2. CAG Repeat Length in Our ATXN1[82Q] Mice Shortened from 82 to 71 Repeats

CAG repeats are unstable and can expand or shrink with each generation. As creating triple transgenic mice requires several generations, to confirm the CAG repeat length in the *ATXN1* gene of our novel *ATXN1[82Q];IKKβ$_{CA}$;Slc1a3-cre/ER$^T$* line, we sequenced the CAG repeat length of the *ATXN1[82Q]* transgene (Figure 2). We found that the CAG repeat length of the triple transgenic mice had shortened to 71 repeats (Figure 2B) from the 82 CAG repeats in our progenitor *ATXN1[82Q]* mice (Figure 2C). Henceforth, we will refer to the mutant *ATXN1* transgene as *ATXN1[71Q]* and our triple transgenic mice as *ATXN1[71Q];IKKβ$_{CA}$;Slc1a3-cre/ER$^T$*.

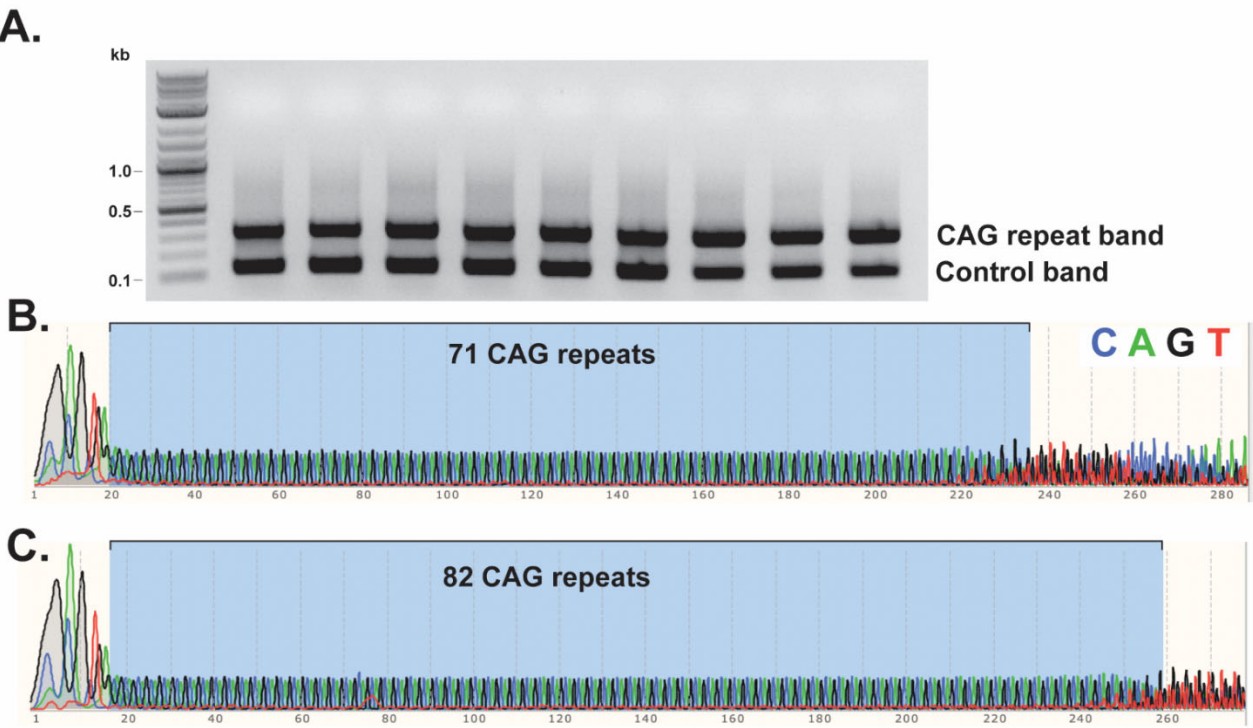

**Figure 2.** Novel SCA1 mice have spontaneous reduction in CAG repeats in *ATXN1[82Q]* transgene. PCR products of the CAG repeat region of the *ATXN1[82Q]* transgene from triple transgenic SCA1 mice on a 1% agarose gel (**A**). Lanes represent different animals, with the upper band showing the PCR product of CAG repeats and the lower band showing the control PCR product. Chromatograms of the CAG repeat sequence from the *ATXN1[82Q]* transgene of novel SCA1 triple transgenic mice (**B**) and *ATXN1[82Q]* progenitor mice (**C**). Chromatograms were made using SnapGene Viewer 6.0 software.

### 3.3. Expression of IKKβCA Does Not Significantly Alter GFAP Reactivity in Bergmann Glia

Increased GFAP immunoreactivity is a commonly used indicator of reactive astrogliosis [20]. We used GFAP immunofluorescence to investigate whether expressing *IKKβ$_{CA}$* in Bergmann glia of SCA1 or wild-type mice impacted this hallmark of astrogliosis (Figure 3A). We found a trending increase in GFAP immunoreactivity of Bergmann glia in *IKKβ$_{CA}$* mice compared to wild-type control mice at 12 weeks. There was no difference between GFAP intensity in Bergmann glia in *ATXN1[71Q]* and *ATXN1[71Q];IKKβ$_{CA}$* mice at

either mid (12 weeks) or late (22–26) stages of disease (Figure 3B, Supplementary Figure S3). These results suggest that further activating NF-κB does not significantly impact cerebellar Bergmann glia reactive gliosis in SCA1 mice.

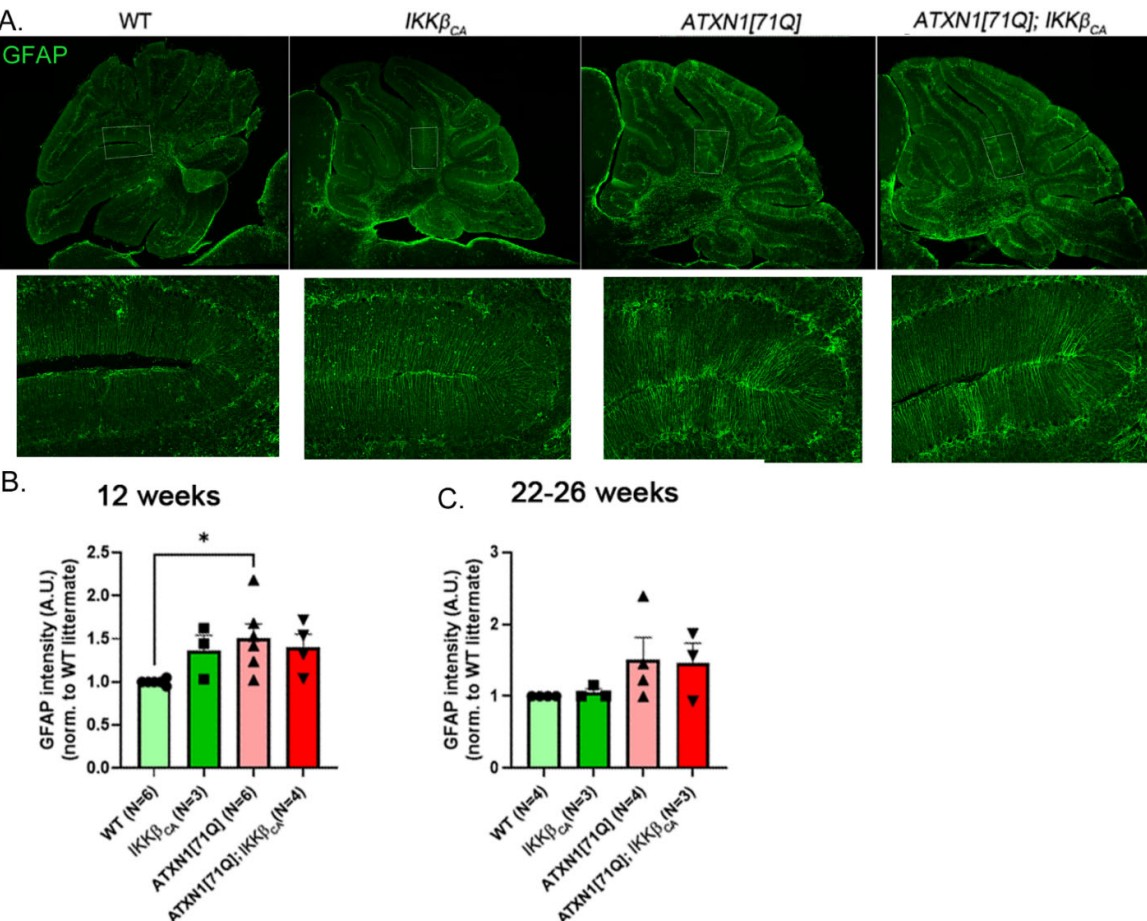

**Figure 3.** Bergmann glia GFAP intensity is increased in SCA1 mice but not significantly altered with *IKKβ$_{CA}$* expression. Representative GFAP immunofluorescence staining of the whole cerebellum of 12-week-old mice. Boxed regions of the molecular layer are shown below (**A**). Image J was used to quantify the intensity of GFAP staining at 12 weeks (**B**) and 22–26 weeks (**C**). Data are presented as mean ± SEM with average values for each mouse represented by a dot * $p < 0.05$ using one-way ANOVA with Bonferroni's multiple comparison test. Error bars = S.E.M.

### 3.4. 71 CAG Repeats Is Not Sufficient to Cause Motor Deficits

SCA1 transgenic mice have been well characterized with a robust onset of motor symptoms by 12 weeks of age. Our previous results indicated that inhibiting early NF-κB-dependent astrogliosis worsens motor deficits in *ATXN1[82Q];IKKβ flox/flox;Slc1a3-cre/ER$^T$* mice, indicating its beneficial effects. To determine whether further activating the astroglial NF-κB pathway can provide additional therapeutic benefits, we examined mouse motor performance throughout disease progression via rotarod at 12, 17, and 22 weeks. We used these ages because *ATXN1[82Q]* mice show robust motor deficits at 12 weeks age [49,50], and it allowed sufficient time (8 weeks) for the mice to recover from IP TMX injections, which might affect rotarod performance, and for an increase in astroglial NF-κB signaling to impact disease. Four groups of mice were tested on the rotarod: WT, *IKKβ$_{CA}$*, *ATXN1[71Q]*, and *ATXN1[71Q];IKKβ$_{CA}$*.

At 12 weeks of age, *IKKβ$_{CA}$* mice had a trending increase in the latency to fall on rotarod compared to wild-type mice (latency for WT mice was 235.3 s, and *IKKβ$_{CA}$* 273.1 s). Similarly, *ATXN1[71Q];IKKβ$_{CA}$* mice stayed longer on a rotarod compared to *ATXN1[71Q]*

mice (latency for *ATXN1[71Q]* mice was 228.6, and *IKKβ*$_{CA}$ 269.8 s) and again this difference was not statistically significant. Perplexingly, despite previous studies, we did not detect a significant decrease in latency to fall in *ATXN1[71Q]* mice compared to wild-type mice at either 12 weeks or at older ages (Figure 4).

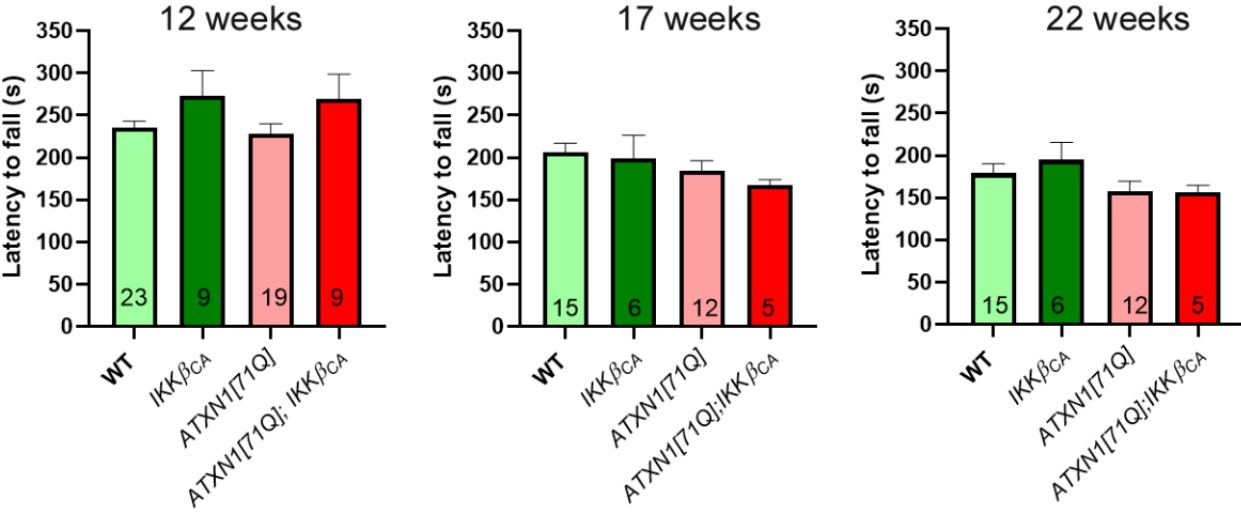

**Figure 4.** Rotarod performance of *ATXN1[71Q]* and *ATXN1[71Q];IKKβ*$_{CA}$ mice is not impaired even at late stages of disease. Rotarod performance of mice at 12, 17, and 22 weeks. Error bars = S.E.M., using one-way ANOVA with Bonferroni's multiple comparison test. Number of mice used for quantification is present in bars.

Because the length of CAG repeats in the *ATXN1* gene correlates with worse symptoms, the reduction in CAG repeat length from 82 to 71 in our colony likely contributed to statistically undistinguishable rotarod performance between WT and *ATXN1[71Q]* mice.

### 3.5. Pathology of Purkinje Neurons Is Still Detectable in ATXN1[71Q] Mice

Next, we determined to what extent a reduction to 71 CAG repeats and activating astroglial NF-κB signaling early in the disease affects Purkinje cell dendrite atrophy and glutamatergic synapse loss, well-characterized hallmarks of SCA1 pathology in *ATXN1[82Q]* mice [32,34,49,53,58,59]. To quantify dendritic atrophy and synaptic loss, we used antibodies against calbindin, a marker of Purkinje neuron soma and processes, and vesicular glutamate transporter 2 (VGLUT2), a marker of climbing fiber synapses onto Purkinje neurons (Figure 5A, Supplementary Figure S4).

At 12 weeks, we found a decrease in the molecular layer width, indicating Purkinje cell atrophy, in *ATXN1[71Q]* mice when compared to wild-type mice (Figure 5B). Importantly, there were no differences in the molecular layer width between *IKKβ*$_{CA}$ and *ATXN1[71Q];IKKβ*$_{CA}$ mice at 12 weeks indicating that atrophy of Purkinje cells is ameliorated when astroglial NF-κB is further increased early in the disease.

However, we found no significant change in the calbindin intensity, or VGLUT2 synapse Purkinje cell dendrites (measured as the length of climbing fiber VGLUT2 positive synaptic terminals on calbindin-labeled Purkinje cell dendrites), in *ATXN1[71Q]* mice when compared to wild-type mice (Figure 5B), indicating that reduction in CAG repeat length from 82 to 71 likely ameliorated these hallmarks of SCA1 pathology. This also precluded evaluation of whether further activating astroglial NF-κB can ameliorate these SCA1 hallmarks.

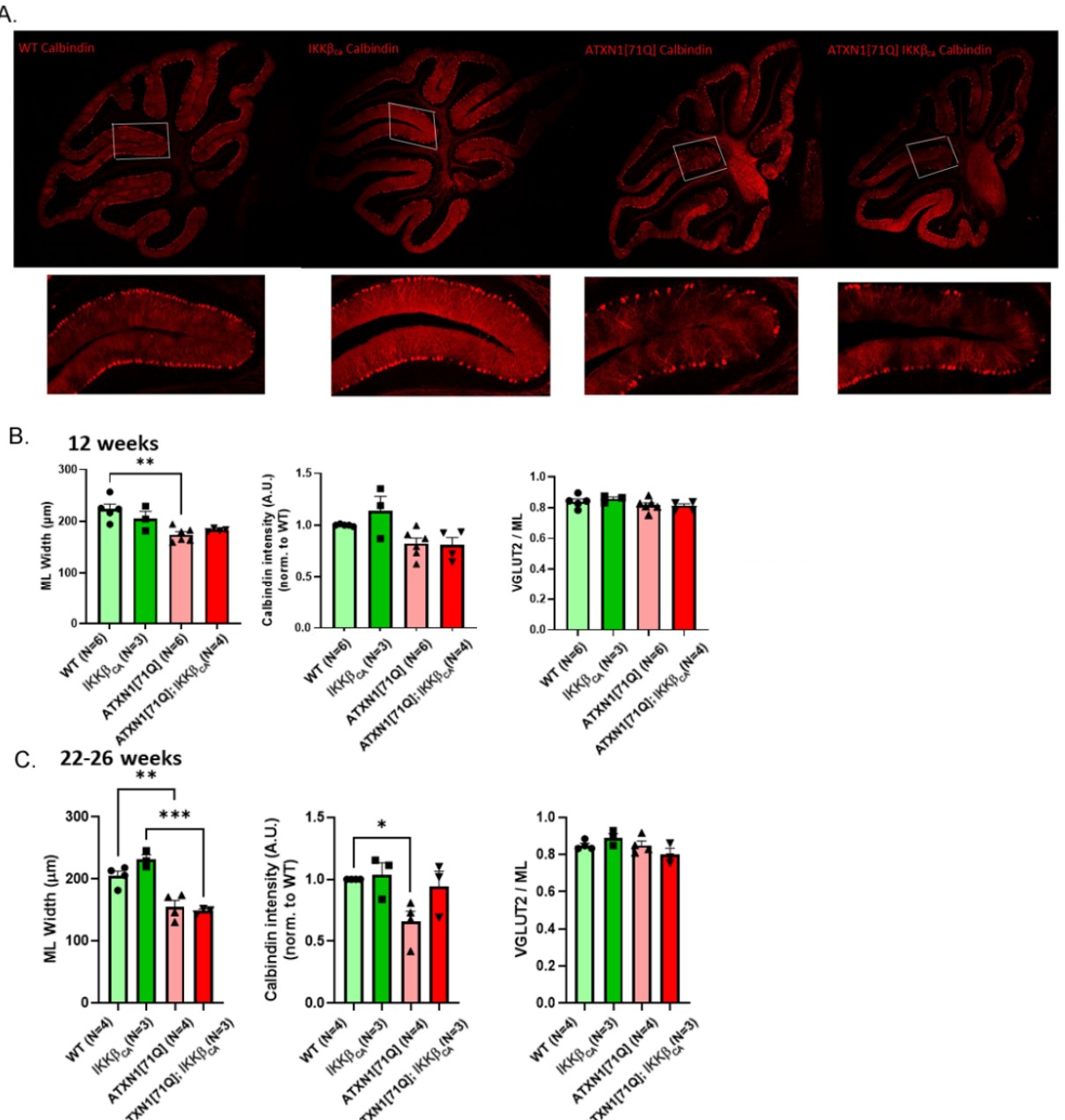

**Figure 5.** *ATXN1[71Q]* mice demonstrate Purkinje cell pathology despite lack of motor phenotype. Representative calbindin immunofluorescence staining of the whole cerebella of 12-week-old mice. Boxed regions of the molecular layer are shown below. (**A**) Cerebellar slices from 12- (**B**) and 22–26-week-old (**C**) mice were stained with antibody specific for Purkinje neuron-marker calbindin or with antibodies against calbindin and vesicular glutamate transporter 2 (VGLUT2) to label climbing fiber synapses on PNs. ImageJ was used to quantify molecular layer thickness, calbindin intensity in the Purkinje neurons, and length of climbing fiber synapses (VGLUT2 puncta) on Purkinje neuron dendrites (VGLUT2/calbindin). Data are presented as mean ± SEM with average values for each mouse represented by a dot. * $p < 0.05$, ** $p < 0.005$, *** $p < 0.0005$ using one-way ANOVA with Bonferroni's multiple comparison test. Error bars = S.E.M.

At 22 weeks, molecular layer width was significantly decreased both in *ATXN1[71Q]* and *ATXN1[71Q];IKKβ_{CA}* mice compared to their respective littermates. However, while calbindin protein expression was now significantly decreased in *ATXN1[71Q]* mice when

compared to wild-type mice, there were no differences in the calbindin expression between *IKKβ$_{CA}$* and *ATXN1[71Q];IKKβ$_{CA}$* mice at 22 weeks. This may indicate delayed or ameliorated Purkinje cell pathology with astroglial *IKKβ$_{CA}$* expression in SCA1 transgenic mice (Figure 5C).

Cerebellar Purkinje cell degeneration is also characterized by decreased expression of a set of genes known to be significantly correlated with disease progression in Purkinje cells (Ingram et al., 2016). Expression of these genes was quantified in the cerebellar RNA extracts of WT, *IKKβ$_{CA}$*, *ATXN1[71Q]*, and *ATXN1[71Q];IKKβ$_{CA}$* mice at 12 and 22–26 weeks of age (Figure 6). We found a significant decrease in the expression of these Purkinje cell genes in *ATXN1[71Q]* mice at both examined time points, despite a lack of motor deficits. We found no difference between *ATXN1[71Q]* and *ATXN1[71Q];IKKβ$_{CA}$* mice in the expression of these genes.

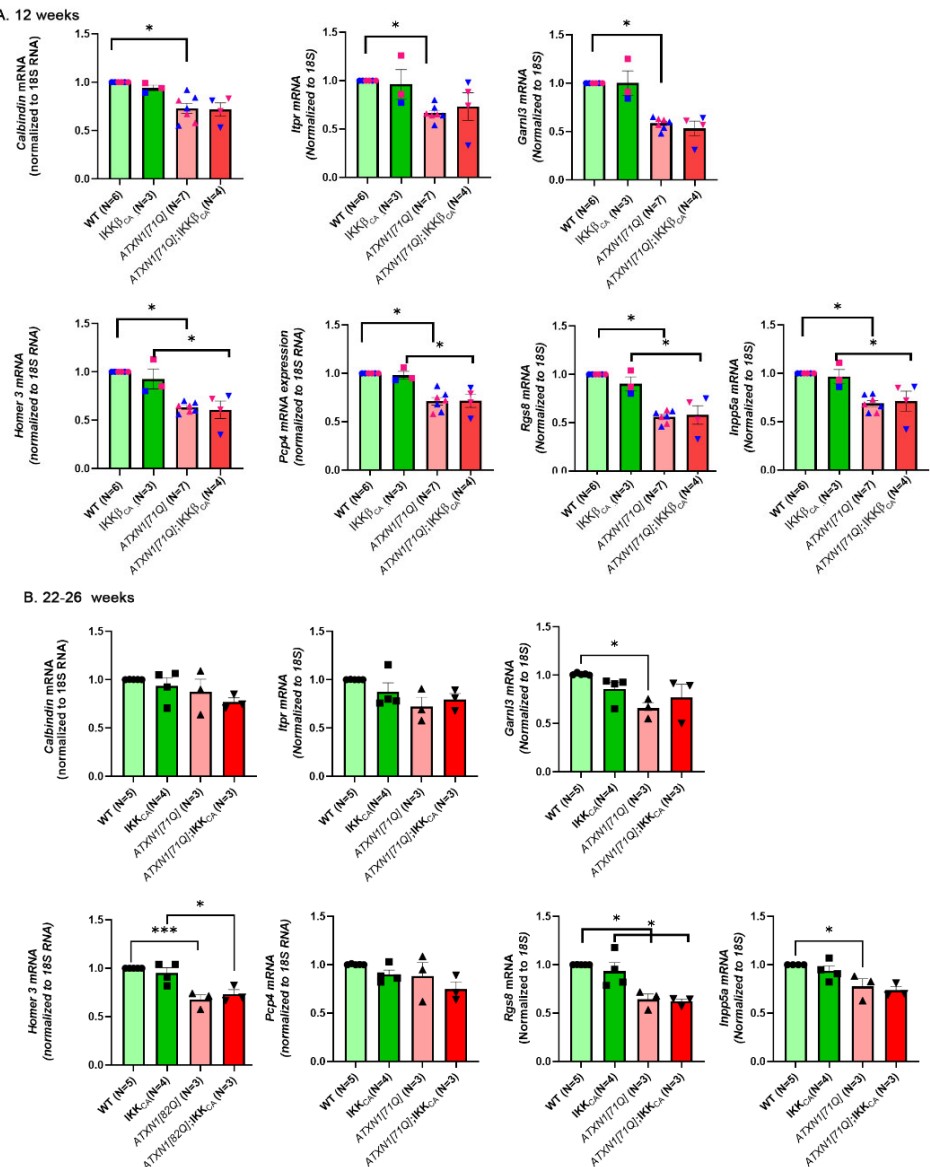

**Figure 6.** Expression of Purkinje cell genes associated with SCA1 disease is decreased in *ATXN1[71Q]* mice. The cerebellum was dissected from 12- and 22–26-week-old mice, mRNA was extracted, and RT-qPCR was used to evaluate expression of Purkinje cell genes associated with SCA1 disease. Data are presented as mean ± SEM with average values for each mouse represented by a dot. * *p* < 0.05, *** *p* < 0.0005 using one-way ANOVA with Bonferroni's multiple comparison test. Error bars = S.E.M.

In *ATXN1[82Q]* cerebella, loss of Purkinje cells occurs around 24 weeks of age [50,53,54]. Therefore, we evaluated Purkinje cell loss at the late disease stage (22–26 weeks). The number of Purkinje cells was not changed in *ATXN1[71Q]* mice compared to wild-type controls, nor in *IKKβ$_{CA}$* lines (Supplementary Figure S5).

Together, these results indicate that loss of 11 CAG repeats in mutant ATXN1 rescues motor phenotype, but Purkinje cell atrophy and molecular changes are still detectable. Moreover, further activating astroglial NF-κB signaling seems to ameliorate and delay atrophy but not gene expression changes in Purkinje cells.

### 3.6. Activation of Astroglial NF-κB Signaling Early in Disease Slightly Increases Microglial Density

Microglia are widely accepted as the resident immune cells of the brain that become reactive upon insult [60,61]. Moreover, there is mounting evidence supporting the importance of astroglia–microglia crosstalk in neurodegenerative diseases [46,62]. One way to assess microglial reactive gliosis is to quantify the increase in microglial density using immunofluorescence with microglial marker ionized calcium-binding adapter molecule 1 (Iba1) (Figure 7A). We previously reported increased microglial density in the cerebella of pre-symptomatic SCA1 mice co-occurring with cerebellar astrogliosis [31]. We found an increased density of microglia in *ATXN1[71Q]* mice (Figure 7), indicating that reduced CAG repeat number can still cause microglial reactive gliosis. We also examined whether further activation of astroglial NF-κB signaling affects microglial activation in SCA1. We observed a slight increase in the density of Iba-1-positive microglia in TMX-treated *ATXN1[71Q];IKKβ$_{CA}$;Slc1a3-cre/ER$^T$* mice both at 12 weeks and at 22–26 weeks, but this did not reach statistical significance (Figure 7B,C). This result suggests that astroglial NF-κB may promote microglial density in SCA1.

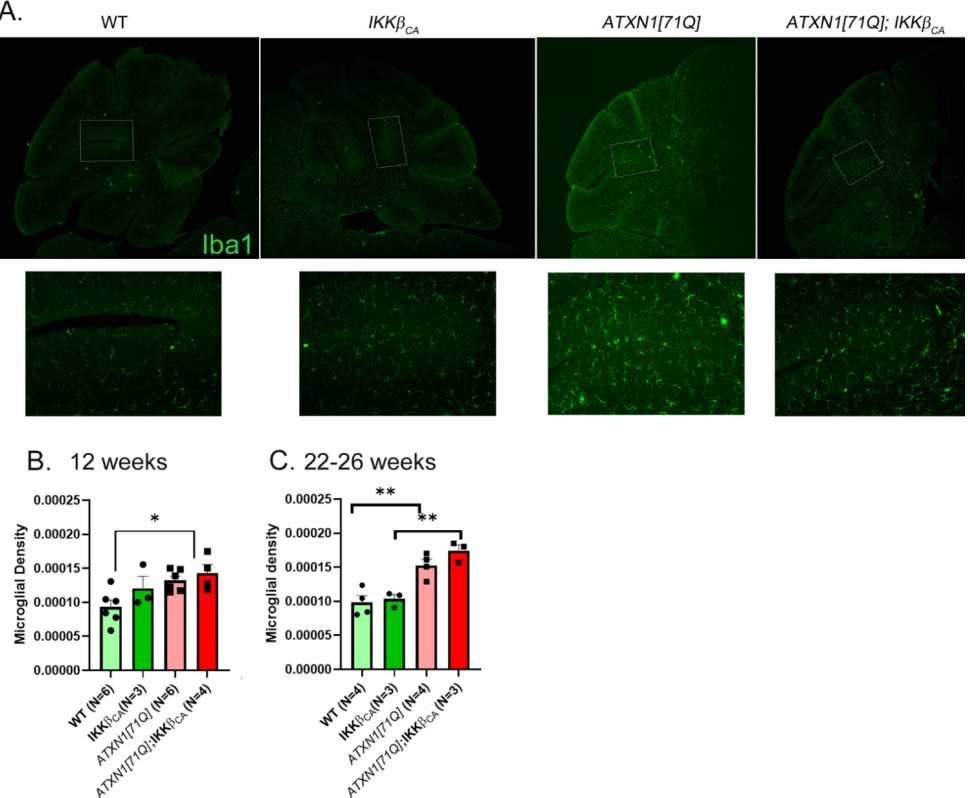

**Figure 7.** Increased microglia density in cerebella of *ATXN1[71Q]* mice. Representative Iba1 immunofluorescence staining of the whole cerebella of 12-week-old mice. Boxed regions of the molecular

layer are shown below. (**A**) Scale bar = 100 μm. Image J was used to quantify the density of Iba1-positive microglia at 12 weeks (**B**) and 22–26 weeks (**C**). Data are presented as mean ± SEM with average values for each mouse represented by a dot. * $p < 0.05$, ** $p < 0.005$ using one-way ANOVA with Bonferroni's multiple comparison test. Error bars = S.E.M.

## 4. Discussion

Astrocytic NF-κB signaling is thought to influence the pathology of several neurodegenerative diseases, including Alzheimer's disease, Huntington's disease, and SCA 1 and 3 [34,36,63]. In SCA1 specifically, activation of astrocytic NF-κB signaling—as evidenced by increased IKKβ and NF-κB phosphorylation—has been noted in cerebellar astrocytes [34]. In the cerebellum, activation of NF-κB in astrocytes was previously shown to be sufficient to trigger ataxia symptoms such as neuroinflammation, astrogliosis, and Purkinje cell loss [64]. However, we did not see these effects, most likely due to lower levels of IKKβ$_{CA}$ activation in the IKKβ$_{CA}$ line we used [55].

Moreover, astrocytic NF-κB signaling can have both beneficial and detrimental effects that depend on the stage of disease [34,65]. For example, in a model of ALS, astrocytic NF-κB activation exerted a neuroprotective effect on motor neurons in the pre-symptomatic phase of the disease, but accelerated disease progression in the later stages of ALS [65]. In SCA1, Kim et al., using IKKβ deletion to inhibit NF-κB demonstrated a biphasic role of astrocytic NF-κB signaling, where NF-κB had a beneficial role in the early stages of SCA1, and the opposite effect was shown in post-symptomatic SCA1 transgenic mice with astrocytic NF-κB inhibition. As the study by Kim et al. showed beneficial effects of astrocytic NF-κB signaling early in SCA1, here we explored whether further activating astrocytic NF-κB signaling by inducing expression of constitutively active IKKβ could provide additional benefits. Our results suggest that astroglial expression of IKKβ$_{CA}$ delays Purkinje cell atrophy and reduced calbindin expression. However, the change in the phenotype of these mice with CAG reduction to 71 repeats prevented us from evaluating the neuroprotective potential of IKKβ$_{CA}$ expression on motor deficits and synaptic loss in these mice.

There are two additional important findings in our study. First, mice in our colony spontaneously decreased the number of CAG repeats from 82 CAG to 71 CAG in the *ATXN1* gene. Impressively, we found that this shrinkage of 11 CAG repeats in *ATXN1* gene is sufficient to cause loss of motor phenotype. However, we were still able to detect significant pathology of Purkinje cells, astrogliosis, and microgliosis. These results indicate that despite detectable dysfunction in key cerebellar cells, motor performance is not impaired in *ATXN1[71Q]* mice. Future studies using electrophysiology will examine the functional impairments of Purkinje cells in these mice. However, there is one SCA1 phenotype we could not detect in *ATXN1[71Q]* mice—it was a loss of climbing fiber synapses on Purkinje cells. Our results suggest that dendritic atrophy and perturbed gene expression in Purkinje cells, astrogliosis, and microgliosis precede loss of synapses. Moreover, our results indicate that loss of climbing fiber synapses correlates well with motor deficits in these mice.

## 5. Main Points

1.  Further activating the NF-κB pathway in astrocytes is not beneficial in Purkinje cell-specific transgenic Spinocerebellar ataxia type 1 mice.
2.  Modest reduction in CAG repeat length rescues motor deficits but not Purkinje cell pathology in Spinocerebellar ataxia type 1 mice.

**Supplementary Materials:** The following supporting information can be downloaded at https://www.mdpi.com/article/10.3390/neuroglia4010005/s1. Figure S1. Rotarod performance of *ATXN1[82Q]* mice is impaired at 12 weeks. Rotarod performance of mice at 12 weeks of age. Data are presented as mean ± SEM with average values for each mouse represented by a dot. * $p < 0.05$ using Student's *t*-test. Error bars = S.E.M. Figure S2. Selective recombination in cerebellar astrocytes in *Slc1a3-CreER$^T$* upon tamoxifen injection. *Slc1a3-CreER$^T$* mice were crossed with reporter mice that

have loxP sites surrounding the STOP codon in front of the TdTomato gene. TMX injection resulted in TdTomato signal in astroglial cells, marked by expression of S100B. Figure S3. Representative GFAP immunofluorescence staining in the cerebellar molecular layer of 22–26-week-old mice. Figure S4. Representative VGLUT2 (green) and calbindin (red) immunofluorescence staining in the cerebellar molecular layer of 12-week-old mice. Figure S5. Purkinje cell number is not altered in *ATXN1[71Q]* mice. Image J was used to quantify the number of calbindin-positive Purkinje cells at 22–26 weeks. Data are presented as mean $\pm$ SEM with average values for each mouse represented by a dot. No statistical differences using one-way ANOVA with Bonferroni's multiple comparison test. Error bars = S.E.M.

**Author Contributions:** M.C. conceived this study. M.C., S.G., J.-G.R., K.S. and G.B. performed the experiments, analyzed the data, prepared the figures, and wrote the manuscript. W.L. provided IKK$_{CA}$ mice. All authors have read and agreed to the published version of the manuscript.

**Funding:** This work was supported by National Institute of Health NINDS awards (R01 NS197387 and R01 NS109077 to M.C.).

**Institutional Review Board Statement:** Animal experimentation was approved by the Institutional Animal Care and Use Committee (IACUC) of University of Minnesota (protocol number 2109-39465A last approval date 02/20/2023) and was conducted in accordance with the National Institutes of Health's Principles of Laboratory Animal Care (86–23, revised 1985), and the American Physiological Society's Guiding Principles in the Use of Animals.

**Informed Consent Statement:** Not applicable.

**Data Availability Statement:** Data are available on request from the authors.

**Acknowledgments:** We are grateful to Orr and Lin for the generous gift of mouse lines and to all the members of the Cvetanovic and Orr laboratories for suggestions.

**Conflicts of Interest:** The authors have no conflict of interest to declare.

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
