# Peer review of "Modest Reduction in CAG Repeat Length Rescues Motor Deficits but Not Purkinje Cell Pathology and Gliosis in Spinocerebellar Ataxia Type 1 Mice"

_2571-6980, doi:10.3390/neuroglia4010005_

Round 1

Reviewer 1 Report

The present article explains how a slight reduction in the number of CAG repeats in the ataxin-1 gene in an animal model of spinocerebellar ataxia type 1 is sufficient to delay motor deficits but not to correct the pathology in Purkinje cells. The article is interesting, however I think that there are some details to be considered:
- In the abstract you said that "further activating NF-kB signaling in Bergman glia in these mice does not provide therapeutic benefits", but I think the phenotype change is an artifact that does not allow you to assess whether early-stage NF-KB is neuroprotective or not. So I would try to explain that the change in the phenotype of these mice has prevented us from evaluating the neuroprotective potential of IKKbca expression in these mice. And you could provide some strategies to fully study the potential of NFkB in this animal model. 

- The figure number 1 has too low quality, and it is difficult to follow it. 

- As there are some drugs that can inhibit NF-KB signaling it would be interesting to know what is happening in this new ATXN1(71) mice. 

Author Response

1.- In the abstract you said that "further activating NF-kB signaling in Bergman glia in these mice does not provide therapeutic benefits", but I think the phenotype change is an artifact that does not allow you to assess whether early-stage NF-KB is neuroprotective or not.

We thank the reviewer for this suggestion and have modified abstract accordingly.

We have found delayed PC atrophy and calbindin reduction in SCA1 mice expressing IKKβCA in astrocytes implicating beneficial effects of increased NF-κB signaling on Purkinje cell pathology. The change in the motor phenotype of SCA1 mice with CAG reduction has prevented us from evaluating the neuroprotective potential of IKKβCA on motor deficits in these mice.

  1. The figure number 1 has too low quality, and it is difficult to follow it. 

We have created a new Figure 1 with increased quality that is hopefully easier to follow.

  1. As there are some drugs that can inhibit NF-KB signaling it would be interesting to know what is happening in this new ATXN1(71) mice. 

This is a very good suggestion. However, as the lack of motor phenotype precludes their assessment of motor phenotype and due to the cost of maintenance we have discontinued ATXN1[71Q] mice in our colony so we cannot try this experiment.

Reviewer 2 Report

In this study the authors analyze NF-κB activation in astrocytes at an early stage of Spinocerebellar ataxia type 1 (SCA1) disease, as possible therapeutic benefits.

They use a triple transgenic mouse line (ATXN1[82Q];IKKβCA;Slc1a3-cre/ERT) for selective activation of classical NF-κB signaling in SCA1 astroglia, starting at 4 weeks of age. They report that during the crossings the CAG repeat number of the ATXN1 gene was decreased from 82 to 71 and clearly show that ATXN1[71Q] mice do not exhibit motor deficits comparable to ATXN1[82Q] mice, although they could still detect significant pathology of Purkinje cells, astrogliosis and microgliosis, but not loss of climbing fiber synapses on Purkinje cells.

In my opinion some points need to be addressed more carefully.

Please display representative images of VGLUT2 staining at 12 or 22-26 months, as only graphs are shown in Fig. 5B-C.

Fig.3A, 5A, 7A: Please use better images: I suggest displaying the entire cerebellum and possibly enlargement from the same lobule in each mouse line.

Methods and Materials section 2.5: describe area from which GFAP staining was quantified.

Fig.4: as the natural decline due to age is not observable in WT or IKKCA mice, it would better to perform other behavioral tests to measure balance and coordination.

Minor points:

Methods and Materials section: please describe the genetic background of the 3 mouse lines.

Suppl. Figure 1 only describe motor coordination deficits, dendritic atrophy of Purkinje neurons is not shown.

Author Response

  1. Please display representative images of VGLUT2 staining at 12 or 22-26 months, as only graphs are shown in Fig. 5B-C.

We thank the reviewer for this comment and have included representative images of VGLUT2 staining in Supplementary Figure 4.

  1. 3A, 5A, 7A: Please use better images: I suggest displaying the entire cerebellum and possibly enlargement from the same lobule in each mouse line.

This is excellent suggestion; we have included new entire cerebellum images in revised Figs 3A 5A and 7A.

  1. Methods and Materials section 2.5: describe area from which GFAP staining was quantified.

We have added this information to Methods and Materials.

The intensity of staining for calbindin and GFAP was determined by measuring mean gray value in the molecular and Purkinje cell layers of lobule VI of the cerebellum. Microglia density was determined by dividing the number of Iba-1-positive microglia found within the molecular and Purkinje cell layers of lobule VI of cerebellar cortex by the region of interest (ROI) area.

  1. 4: as the natural decline due to age is not observable in WT or IKKCA mice, it would better to perform other behavioral tests to measure balance and coordination.

We thank the reviewer for this insightful comment. Rotarod graphs were made by Prizm automatically determining the range of Y axes so the natural decline due to age change was not visible. In revised Figure 4 we have made ranges of Y axes consistent among different age graphs so that decline can be visible (i.e. in WT mice average latency to fall was 235.3, 206.4, and 179.1 at 12, 17 and 22 weeks respectively).

Minor points:

  1. Methods and Materials section: please describe the genetic background of the 3 mouse lines.

We thank the reviewer for this suggestion, revised methods describe C57B6/L as genetic background of all of the mouse lines.

All of the mouse lines are on C57BL/6 background. Creation and the genetic background of the ATXN1[82Q] mice has been previously described (Burright et al. 1995). While these mice were originally generated in FVB/N background, we have backcrossed them into C57BL/6 mice for 10 generations. In this mouse model, the expression of mutant ATXN1 with an expanded 82 CAG repeat in its coding sequence, is driven by a Purkinje cell specific 2(Pcp2/L7) promoter. We received these mice as a gift from Dr. Harry Orr. In order to achieve selective activation of astroglial NF-κB signaling, ATXN1[82Q] mice were crossed with R26StopFLIKKβCA mice on C57BL/6 background, received as a gift from Dr. Wensheng Lin (Lei et al. 2020). These mice feature a constitutively active form of IKKβ, in which activation is mimicked through two serine to glutamine substitutions in the activation loop of the kinase domain (Sasaki et al. 2006). Expression of the IKKβCA transgene, its FLAG tag, and a GFP reporter is blocked by a floxed STOP cassette (Lei et al., 2020). When bred to mice expressing tamoxifen-activated CreERT recombinase under the control of the GLAST/Slc1a3 promoter (Jackson laboratory, #012586, Tg(Slc1a3-cre/ERT)1Nat/J)(Kim et al. 2018), following tamoxifen injection, the floxed STOP cassette is deleted and astrocyte-specific IKKβCA expression is achieved. Slc1a3-cre/ERT mice on C57BL/6 background were obtained from the Jackson laboratory (#012586; Tg(Slc1a3-cre/ERT)1Nat/J).

  1. Figure 1 only describe motor coordination deficits, dendritic atrophy of Purkinje neurons is not shown.

We have included several references that have previously examined dendritic atrophy in ATXN1[82Q] mice including studies from our group.

We next determined to what extent a reduction to 71 CAG repeats and activating astroglial NF-κB signaling early in disease affects Purkinje cell dendrite atrophy and glutamatergic synapse loss, well-characterized hallmarks of SCA1 pathology in ATXN1[82Q] mice (Duvick et al. 2010)(Gennarino et al. 2015)(Ferro et al. 2018)(Qu et al. 2017)(Kim et al. 2018)(Mellesmoen et al. 2019).

Reviewer 3 Report

Gilliat S., et al have evaluated that pro-inflammatory transcriptional regulator nuclear factor κ-light-chain-enhancer of activated B cells (NF-κB) signaling in astrocytes has a neuroprotective role during an early stage of Spinocerebellar ataxia type 1 (SCA1), a dominant inherited neurodegenerative disease caused by the expansion of CAG repeats in the Ataxin-1 (ATXN1) gene, using a novel Slc1a3-CreERT/IKKβCA/ATXN1[82Q] triple transgenic mouse model.

 In this study, the authors managed to demonstrate that ATXN1[82Q] mice no longer exhibit motor deficits at very late stages of the disease. But mice showed fewer CAG repeats in the ATXN1 gene, which eventually cause loss of motor phenotype. Importantly, they found crucial pathologies of Purkinje cells, as well as astrogliosis and microgliosis, which suggests that motor performance is not impaired in ATXN1[71Q] mice, which have subsequent defective cerebellar cells.

 Although these results conclude a novelty towards therapeutic benefits of motor dysfunction in SCA1 but one of the critical phenotypes, loss of climbing fiber synapses was not exhibited by ATXN1[71Q] mice and IKKβCA expression does not make any difference in SCA1-pathogenesis. But needs further region and cell-specific validation in mice.

 The experiments designed for this study are justified but a few more supporting protein expression profiles would provide more significance to this study. Overall, the results are significant but there is a lack of experimental data to justify the conclusions which need appropriate controls to be performed to validate the importance of CAG repeats. The authors also have missed mentioning supporting references for SCA1 and the importance of CAG repeats.

Nonetheless, the article seemed to possess a few major concerns related to the pathology of Purkinje cells in SCA1. The novelty of this study is shrinkage of 11 CAG repeats in the ATXN1 gene is sufficient to cause loss of motor phenotype, which poses a crucial therapeutic target in SCA1. The authors should use low mag fluorescence microscopy images to show the Pathology of Purkinje neurons in the cerebellum. This data will give us another set of valuable information on overall structural changes in WT vs mutant mice models, which could be highly relevant to SCA1.

Overall, the clarity of the text is good but needs some readjustments. The manuscript has very few typographical and grammatical errors. A few main figures require a bit of attention on the color code of the graphs. The quantitative analyses are much appreciated. The authors need to describe some of the results with supplementary data. In general, the manuscript can accomplish the caliber of quality for consideration for publication in the Neuroglia journal with some details. The authors are advised to consider the comments below:

Major comments

1.      Result /3.1. The creation of mice with conditional, astroglia-selective, and TMX-dependent expression of constitutively active IKKβ/ Needed control experiment provides evidence that IKKbCA is only expressed in astrocytes, not in any neurons.

2.      Please provide Representative GFAP immunofluorescence staining in the cerebellar molecular layer of 22-26-week-old mice (as supplementary data). One of the important measurements of protein expression is via western blots. It would be helpful to see immunoblots of GFAP expression from specific brain regions like the cerebellum of SCA1 mice.

3.      Result / 3.4.71. CAG repeats is not sufficient to cause motor deficits / One of the important control is missing ATXN1[82Q] mice to support the result. Support from previous literature (as references) also helps to understand the function of CAG repeat.

4.      Please provide representative VGLUT2 immunofluorescence staining as supplementary data.

5.      Result / 3.5. Pathology of Purkinje neurons is still detectable in ATXN1[71Q] mice/ Additional quantification of Purkinje cell number in IKKβCA and ATXN1[71Q]; IKKβCA mice would be highly appreciated.

6.      Immunofluorescence staining of Homer3 in Purkinje neurons would provide more evidence for the pathology of Purkinje cells.

Minor comments

1.      Please be consistent with writing – main figures mention as “Figure 1,2 …” make sure to use the same proforma in Supplementary.

2.      Please indicate the ladder markers in Figure 2A.

3.      Spelling check of “ IKKCA “ in the Figures throughout the manuscript.

4.      Please check the color code of the bar graph in Figure 4. A consistent color code is necessary for better representation.

Author Response

Major comments

  1. Result /3.1. The creation of mice with conditional, astroglia-selective, and TMX-dependent expression of constitutively active IKKβ/ Needed control experiment provides evidence that IKKbCA is only expressed in astrocytes, not in any neurons.

We thank the reviewer for this suggestion. To address this comment in the revised manuscript we have included new Supplementary Figure 2 in which using reporter mice, we demonstrate selective and almost 100% recombination in cerebellar astrocytes. We have also tried to obtain data showing astrocyte selective increase in IKKCA expression. However, we were not able to obtain good staining with IKKCA antibody we used.

  1. Please provide representative GFAP immunofluorescence staining in the cerebellar molecular layer of 22-26-week-old mice (as supplementary data).

We have included representative GFAP immunofluorescence staining in the cerebellar molecular layer of 22-26-week-old mice as supplementary Fig 4 in the revised manuscript.

  1. Result / 3.4. CAG repeats is not sufficient to cause motor deficits / One of the important control is missing ATXN1[82Q] mice to support the result. Support from previous literature (as references) also helps to understand the function of CAG repeat.

We thank the reviewer for this suggestion. Supplementary Figure 1 shows motor deficits in ATXN1[82Q] mice and we have also cited previous literature showing motor deficits in these mice.

Motor coordination deficits and dendritic atrophy of Purkinje neurons begin around 12 weeks of age in ATXN1[82Q] mice Supplementary Figure 1)(Clark et al. 1997)(Serra et al. 2006)(Xia et al. 2004)(Ferro et al. 2018b), with loss of Purkinje neurons following around 24 weeks (Clark et al. 1997)(Duvick et al. 2010)(Klement et al. 1998). We previously demonstrated that cerebellar reactive astrogliosis is present in ATXN1[82Q] mice at three weeks of age and thus precedes motor deficits (Cvetanovic et al. 2015).

  1. Please provide representative VGLUT2 immunofluorescence staining as supplementary data.

We have included representative VGLUT2 immunofluorescence staining as Supplementary Figure 4 in the revised manuscript.

  1. Result / 3.5. Pathology of Purkinje neurons is still detectable in ATXN1[71Q] mice/ Additional quantification of Purkinje cell number in IKKβCA and ATXN1[71Q]; IKKβCA mice would be highly appreciated.

We have performed this quantification, it is included as Supplementary Figure 5. As even in ATXN1[82Q] line there is no significant loss until 24 weeks of age, we have not detected any change in the Purkinje cell number in any of the lines.

In ATXN1[82Q] cerebella, loss of Purkinje cells occurs around 24 weeks of age (Clark et al. 1997)(Duvick et al. 2010)(Klement et al. 1998). Therefore, we evaluated Purkinje cell loss at the late disease stage (22-26 weeks). Number of Purkinje cells was not changed in ATXN1[71Q] mice compared to wild-type controls, nor in IKKβCA lines (Supplementary Figure 5)

  1. Immunofluorescence staining of Homer3 in Purkinje neurons would provide more evidence for the pathology of Purkinje cells.

This is a very good suggestion as study by Ruegsegger C et al showed that expression of Homer-3 is decreased and influences SCA1 pathophysiology in a knock-in mouse model of SCA1, Atxn1154Q/2Q mice (Neuron. 2016 ;89(1):129-46). However, while Homer3 mRNA is decreased in ATXN1[82Q], it still remains to be established whether Homer-3 protein levels are also decreased in this transgenic SCA1 line as shown in knock-in Atxn1154Q/2Q mice. We hope that reviewer will agree that this is beyond the scope of our manuscript and will find decrease in Homer-3 mRNA shown in Figure 6 sufficient.

Minor comments

  1. Please be consistent with writing – main figures mention as “Figure 1,2 …” make sure to use the same proforma in Supplementary.

We have corrected this in the revised manuscript.

  1. Please indicate the ladder markers in Figure 2A.

Ladder markers are indicated in the revised Figure 2A.

  1. Please check the color code of the bar graph in Figure 4. A consistent color code is necessary for better representation.

We have tried to match the color code in Figure 4A as best as possible.

Round 2

Reviewer 2 Report

The revised manuscript has been definetively improved and therefore is acceptable for publication.